# The Required Competencies of Bachelor- and Master-Educated Nurses in Facilitating the Development of an Effective Workplace Culture in Nursing Homes: An Integrative Review

**DOI:** 10.3390/ijerph191912324

**Published:** 2022-09-28

**Authors:** Rachida Handor, Anke Persoon, Famke van Lieshout, Marleen Lovink, Hester Vermeulen

**Affiliations:** 1Department of Primary and Community Care, Radboud Institute for Health Sciences, Radboud University Medical Center, 6500 HB Nijmegen, The Netherlands; 2Department of People and Health Studies, Fontys University of Applied Sciences, 5600 AH Eindhoven, The Netherlands; 3Scientific Center for Quality of Healthcare (IQ Healthcare), Radboud University Medical Center, Radboud Institute for Health Sciences, 6500 HB Nijmegen, The Netherlands; 4School of Health Studies, HAN University of Applied Sciences, 6525 EN Nijmegen, The Netherlands

**Keywords:** bachelor nurse, master nurse practitioner, competency, leadership, facilitating, workplace culture, work environment, nursing homes, evidence-based practice, person-centered practice, integrative review

## Abstract

Background: Nursing home care is undergoing significant changes. This requires innovative teams operating in an effective workplace culture characterized by person-centeredness and offering evidence-based care. A pivotal role for bachelor- and master-educated nurses (BNs/MNs) is foreseen to facilitate such cultures; however, there is currently no comprehensive overview of what competencies this requires. Objectives: To identify what competencies are required from BNs/MNs in facilitating the development of an effective workplace culture in nursing homes. Methods and design: We conducted an integrative review (IR) using Whittemore and Knafl’s method. We searched the PubMed, CINAHL, and PsycINFO databases for studies published between January 2010 and December 2021 in English. Two independent reviewers determined whether studies met inclusion: bachelor- or master-educated nurse; nursing home; professional competencies; and mixed methods or qualitative and qualitative studies. We applied the CASP appraisal tool and analyzed the data by applying content analysis. Results: Sixteen articles were included. Five themes were identified representing required competencies for BNs/MNs facilitating: (1) learning cultures in nursing practice; (2) effective work relationships within teams; (3) leadership capability within teams; (4) implementation of guidelines, standards, and protocols; (5) a work environment acknowledging grief and loss of residents within teams. Conclusions: It shows that the BN/MN applies five competencies associated with a facilitator role to promote the development of an effective workplace culture to achieve a safe, high-level quality of care, satisfaction, and well-being. An overarching leadership as a change champion will support teams to achieve a quality that should guide the transformation in nursing care.

## 1. Introduction

Older people nursing needs substantial attention because of increasing multimorbidity and high complexity of care. Several studies have reported that multimorbidity in older people is associated with increased healthcare expenditure and cost [1,2]. Internationally, long-term care for older people has been undergoing a significant change. The nursing home population is changing: nowadays, on average, people enter nursing homes at age 85 or older with five or more morbidities, take nine or more prescription medications, and experience poor cognition and extensive limitations in activities of daily living (ADLs) [3]. Additionally, the proportion of older people has increased and is expected to reach 21% by 2050 [4]. Understanding these trends is critical for projecting the future demands for nursing home care.

At the same time, the institutional model of care is shifting towards humanistic, person-centered care. The care available in the current system does not meet the residents’ individual (with multimorbidity and functional limitations) health-care needs and preferences. A person-centered approach to older people care is necessary to optimally tailor residents’ needs and preferences, and quality and coordination of care [5]. Staff who describe working in a person-centered way elaborate on how they focus on meeting each resident’s needs and improving each resident’s well-being [6]. Person-centeredness is an important condition of high-quality health care. It humanizes health-care delivery and puts older people and their caregivers at the center of their context [7,8,9,10]. Person-centered care emphasizes patient–health-care partnerships, empowering older people to consider their own needs and preferences in the care context, and the need for autonomous decision-making in residential settings. Person-centered care makes an essential contribution to helping older people to maintain their personhood.

Furthermore, a person-centered approach allows healthful relationships to grow and flourish between older people, caregivers, and professionals, creating satisfaction and well-being [7]. Previous studies have reported associations between person-centered care and well-being, thriving, and satisfaction among people who reside in nursing homes, but further evidence is needed [11]. This shift toward person-centered care calls for the development of teams able to facilitate partnerships that enhance the personhood of older people and others significant to them [12]. Person-centered care has also been linked to the well-being and job satisfaction of staff and others. In Vassbo et al.’s [13] study, nurses described working in a person-centered way while thriving at work in terms of a psychological state in which the individuals experience personal growth and development [13]. Nursing teams must be able to respond to these global challenges in health care by ensuring sustainable transformation through practice development methodologies or strategies for change, by maintaining high standards of care for older people, and by improving staff satisfaction in these complex and dynamic contexts [13,14].

Current challenges in older people care require an increased level of professional development for individual nurses, and that they work in teams [15]. This requires an effective workplace culture that is person-centered and offers safe and effective evidence-based health care [16,17]. An effective workplace culture is one that supports knowledge utilization, knowledge transfer, and evidence development. It is recognizable by its shared values, principles, and explicitly made practice standards that culminate in a common vision. An effective workplace culture is enabled by staff who feel empowered and valued for their contributions and included in transparent decision-making for change [18]. Wei et al. [19], McCormack et al. [16] and Manley et al. [17] characterize an effective workplace culture as a positive work environment [16,17,19]. According to Ulrich et al. [20] and Bourgault and Goforth [21], a positive work environment is established through consistent and authentic behaviors: skilled communication, true collaboration, effective decision-making, appropriate staffing, meaningful recognition, and authentic leadership [20,21].

For health-care professionals, a positive work environment is important for good patient care and is strongly associated with attracting and retaining employees, which is crucial in times of health-care staff shortages, especially in older people care. The work environment influences staff psychological safety and satisfaction in nursing homes. Various studies argue that a positive work environment is the key to higher job satisfaction, and that it is crucial to providing high-quality person-centered care in long-term care facilities [22,23]. Among many other benefits, person-centered care improves the staff’s capacity to meet the individual needs of residents with respect. Ineffective workplace cultures have resulted in negative outcomes for patients and have negatively impacted staff well-being [19,24]. Thus, to enable the best outcomes, it is essential to develop an effective workplace culture in which residents and teams interact in a person-centered way and team members’ personal and professional growth and well-being are maximized. To achieve this, Cardiff et al. [25] emphasize the meaning and urgency of transformation in the health care context and the skills required for a person-centered approach. Transformation requires shared values about person-centered care, delivery of care that is both safe and effective, and the embracing of continuous learning, improvement, and development [25]. Furthermore, it requires improving implementation of effective nursing practices.

The facilitator plays a crucial role in enabling and sustaining an effective workplace culture in older people care. A facilitator is an individual skilled in working with the concepts of change management and individual and organizational development [17,26,27]. Facilitation involves working with individuals, teams, and organizations to support them through a successful implementation process [27,28,29]. Nurses with bachelor’s degrees (BNs) and nurses with master’s degrees (MNs) can greatly contribute to the development of the workplace culture and, in doing so, be facilitators in its transformation [30]. A study by White et al. [31] found that nursing leaders can make evidence-based modifications to the organizational processes and culture to support their nurses and certified nursing assistants in delivering safe and effective care. This includes involving direct-care staff in shared decision-making, fostering strong nurse leaders, maintaining evidence-based nursing-care standards, providing opportunities and professional growth for staff, and supporting interdisciplinary teamwork [31]. As Hardy et al. [18] describe, leaders can take on a facilitator to support empowerment processes and thereby enable both the transfer of knowledge into practice and the effective use of the team’s resources [18]. However, the facilitator role is not limited only to leaders. It could also be taken on by BNs/MNs, since they work with many different people, and since they both create culture and are influenced by it. BNs/MNs are ideally positioned and adequately suited to take up the role of facilitator in creating positive work environments through cultural change [30].

A bachelor-educated nurse (BN) is a registered (specialized) nurse with a baccalaureate degree, registration level 6 for health care professionals, conforming to the European Qualifications Framework (EQF). Master-educated nurses (MNs) have completed either an academic degree in nursing science, or a two-year Master of Advanced Nursing Practice (MANP) degree, after which they can register as nurse practitioners (NPs)—a legally protected title in the national registry of nurse practitioners, classified as a EQF level 7 [32]. Although BNs/MNs often actively facilitate in their roles, they are not always specifically recognized as ‘facilitators’ of developing workplace culture and/or positive work environments. Several studies describe the role of BNs/MNs as fostering evidence-based culture in general [33,34]. For example, in Lovink et al. [35], nursing teams in nursing homes described higher-educated nurses as inspiring and supportive facilitators in providing education and collaboration in evidence-based nursing culture [35]. However, the BN/MN is inclined to operate primarily within the clinical patient care pillar of their role. Thus, previous studies have mostly focused on their influence on patient outcomes (microsystems level) rather than on the organizational level.

Very little is known about the competencies required of BNs/MNs in relation to their contribution to developing workplace culture. The current literature contains very few empirical studies about the specific competencies needed for facilitating the development of an effective workplace culture in the nursing home setting. Attention has been focused instead on supporting the empowerment of individual team members in growth and self-development [36,37]. To function in high-complexity situations in older people care, nurses require strong collaboration, communication, reflection, and facilitation skills [38]. More specific competencies for facilitating the development of workplace culture are needed to maintain professional goals. As members of health-care teams, BNs/MNs are positioned to contribute to and lead transformative changes in older people care. The shift to more person-centered and evidence-based older people care requires the identification of a new or enhanced set of competencies [39]. In summary, in this review it is essential to analyze how BNs/MNs attempt to develop culture by ensuring high performance within a highly dynamic, challenging, and complex work environment with unpredictable outcomes. If these competencies are known, they can be systemically developed and strengthened during education, training, and workplace experience.

This review aims to address the following research question: What competencies are required for BNs/MNs in the nursing homes to facilitate the development of an effective workplace culture and thereby contribute to person-centered and evidence-based practice?

## 2. Materials and Methods

### 2.1. Design

Following the five-stage methodology by Whitmore and Knafl, we conducted an integrative review, including research studies with diverse methodologies. The inclusion of a variety of literature sources allowed the concept of BNs/MNs competencies enabling an effective workplace culture in nursing homes to be fully explored. This allows for a more comprehensive understanding of the competencies required of bachelor’s- or master’s-educated nurses employed in nursing homes. It also contributes to theory development processes, presents the standing of science related to the issue, and helps apply this information to practice, policy, and education [40].

### 2.2. Search Method

We explored the PubMed, CINAHL, and PsycINFO databases, restricting the search to studies published in English (see Appendix A). The central keywords and associated synonyms were related to “professional competencies” and the “role of a bachelor nurse” or “master educated nurse” working in a “nursing home setting”, achieving an “effective workplace” or “workplace facilitation of learning.” The search was limited to studies published between January 2010 and December 2021 in English, and the search was conducted between October 2021 and December 2021. The search strategy was designed and conducted with the help of a clinical librarian. Articles were eligible if they explicitly described professional competencies—namely, competencies related to knowledge, skills, and personal social and/or methodological abilities—in work or learning situations, in both professional and personal development.

The inclusion and exclusion criteria are described in Table 1.

### 2.3. Search Outcome

The search resulted in 2646 records (see flow diagram Figure 1) [41]. After removing duplicates, the remaining 1978 were screened on title and abstract by two independent reviewers (RH and ML). If articles met the inclusion criteria, full-text versions of the articles were obtained and further screened for eligibility by two independent reviewers (RH and ML). During the search and selection of eligible articles, the reviewers (RH and ML) noticed a challenge by combining the keywords “Bachelor”, “Master”, and “Advanced” with “nurse(s)” and “nursing”. We also retrieved articles with the search term “registered nurses”. Within the articles it was not always clear which nursing staff members were included. The description or grades of participating nursing staff is unclear or insufficient. The first author ran a second search adding the keyword “registered” to check if the search resulted in more than 2646 records. This was not the case.

AP became involved in case of any disagreement. Consensus was reached through discussion. As a result, 16 articles were eventually included in this review.

### 2.4. Quality Appraisal

Two independent reviewers (RH and AP) conducted the appraisal on all 16 studies using the Critical Appraisal Skills Program (CASP) [42], a critical appraisal tool consisting of different formats for different research designs. We used formats for qualitative and mixed methods studies. The CASP Qualitative Checklist begins with two screening questions (Section A) which can be answered quickly. If the answer to both questions is “yes”, then the user may proceed to the remaining questions in Sections B and C. The key author (RH) evaluated all included articles. The second author (AP) independently evaluated all included articles as well. Discrepancies were resolved by discussion until consensus was reached. A table was developed to provide an overview of the included studies, structured by the relevant CASP questions, and presented with a traffic light legend (see Figure 2).

Many studies received green lights for all questions on the CASP tool. In the CASP qualitative assessment, 10 studies were green in almost all categories. The other qualitative studies were often rated red, as information was lacking about the way in which data collection addressed the research objective, the relationship between researcher and participants, and the recruitment strategy. For the CASP-mixed-method assessment, only one study was red in three categories. The results of this study cannot be transferred to other populations.

### 2.5. Data Analysis

The key author (RH) extracted relevant characteristics from each study (see Table 2), including: title, author, year of publication, aim, country, methodology, setting, and educational level.

The selected studies were uploaded to Atlas.ti 22.0.11 and synthesized using content analysis [58]. Each identified meaning unit was labeled with an open code inductively by two independent reviewers (RH and AP). After five papers’ open coding, we progressed to axial coding. The key author checked that all aspects of the content had been covered by re-reading the articles several times. In the categorization process, categories and themes were identified. Once the categories were established, the key author found the essence of the studied categories [58].

The first five studies were analyzed by two reviewers (RH and AP), and agreement was sought on the open codes and themes through discussion and negotiation. The results that emerged were discussed, defined, and refined by the other authors (FvL, HV, and ML) in three meetings. This comprehensive analysis resulted in five core competencies.

## 3. Results

### 3.1. Description of Included Studies

Sixteen articles were included in the review: four qualitative studies, three case studies, three surveys, two implementation studies, two mixed-methods studies, one action research study, and a consensus study (see Table 2). Six studies used semi-structured interviews, four studies used cross-sectional surveys, one study was an expert consensus study, two studies were implementation studies, and three studies used a combination of qualitative and quantitative processes (see Table 2). Seven studies were conducted in Canada, and the others were carried out in the Netherlands, the United States, the United Kingdom, and Norway. One is a(n) (international) study about the International Consortium on Professional Nursing Practice in Long-Term Care Homes, and improving the professional role of the RN through sharing across nations and identifying the evidence that will support recommendations for improvements in RN leadership. The study samples included five studies about BNs, ten studies about MNs and advanced geriatric nurses, and two studies about RNs, for whom the level of education was unknown.

Eleven studies described competencies regarding facilitating team reflection on care practice, nine studies described competencies regarding educating and training staff at formal moments, and ten studies described competencies regarding educating and training on the job. Fourteen studies described competencies regarding facilitating an effective interprofessional relationship among team members. Out of 16 studies, 13 described competencies regarding facilitating team members into a collective group that provides quality improvement in older people care and creates a culture of engagement that promotes staff autonomy, growth, and well-being. Out of 16 studies, 9 mentioned the facilitating role of a BN/MN in the implementation of guidelines, standards, and protocols. Only 2 out of 16 studies mentioned the supportive role of the BN/MN in the grieving processes.

### 3.2. Competencies

In this review, we identified five themes representing the competencies of the BN/MN in facilitating the development of the nursing home workplace culture: (1) facilitating learning cultures in nursing practice; (2) facilitating effective work relationships within teams; (3) facilitating leadership capability within the team; (4) facilitating the implementation process of guidelines, standards, and protocols; and (5) facilitating a work environment to acknowledge grief and loss of residents within teams.

#### 3.2.1. Facilitating Learning Cultures in Nursing Practice

##### Facilitating Team Reflection on Care Practice

The BNs/MNs coached the team in reflecting on their own professional actions and stimulated the team in reasoning and substantiating choices. The BN/MN fulfilled both a supportive and a communicative role among team members. The BN/MN discussed the challenges the teams were encountering.

“*It [the reflective debriefing groups, led by a nurse specialist] gave scope for reflection on practice and provided a safe environment for staff to make their views known*”.[43]

“*Because there was this sense of analyzing and reflecting back on a resident’s death, staff in the groups (including myself [NP]) were gradually beginning to critically analyze a number of assumptions and issues about end-of-life care*”.[43]

##### Educating and Training Staff at Formal Moments

All the studies identified the BNs/MNs’ role as educator and coach for nursing home staff. The BNs/MNs organized and facilitated formal scheduled sessions to teach and coach the team, used group education sessions and one-on-one educational outreach, and brought the team members together to speak to each other. The BN/MN used a variety of creative teaching strategies.

“*During class sessions [heart failure education], oral and written quizzes were used periodically to stimulate discussion, generate interest, and reinforce learning*”.[44]

“*The nurse practitioner developed a case study (for discussion at a pain team meeting) to work through pain protocol and enhance application of knowledge and problem-solving ability*”.[45]

##### Educating and Training Staff on the Job

The BN/MN was also identified in all the studies as teacher, mentor, and resource person in providing knowledge and skills. The learning strategy of the BN/MN was to start by demonstrating the new skill or service, then coach others and eventually transfer the responsibility. The BN/MN created learning situations in the professional practice to learn with and from each other. The team was facilitated in actively processing information and learning and in providing opportunities for positive feedback.

“*On-the-job skill competency evaluation was conducted by the Geriatric Advanced Practice Nurse to verify learning, answer questions, review Heart Failure worksheets and provide affirmation and positive feedback*”.[44]

“*The effect it has when the BN and direct care staff members look together at the client: ’What do you see? What do I see?’ and that you talk about it. Taking five minutes of your time for that [Board member, organization A, respondent 4]*”.[46]

#### 3.2.2. Facilitating Effective Work Relationships within Teams

The BN/MN role of communicator and collaborator included the whole staff so that each team member was doing his or her own task and working towards the same goal together. The BN/MN stimulated the input of each team member so that staff could come up with their own solutions. The BN/MN created a positive, effective working relationship and collaboration, with and within the team. He or she was active in team building by improving the collaboration processes. This involved creating a powerful connection between team members and a culture where active interaction took place. Active interaction meant that the whole team was involved in the collaboration process and that team members flourished within their own role and position in the team. In addition, it was essential that the team worked towards one common goal with residents and families and with team members, both inside and outside the nursing home. The BN/MN enhanced communication and promoted teamwork [45].

“*According to staff, the collaborative environment and the quality of relationships remained constant regardless of age or seniority, and they [nurse supervisor] facilitated the integration of new nurse and personal support workers [PSW] hires*”.[47]

“*Nursing and interprofessional team members described how the nurse practitioner ‘includes the whole team’, so that each team member was ‘doing [their] piece and all working towards the same goals*”.[48]

“*Participants spoke about how the type of relationship they had with the nurse practitioners or clinical nurse specialist facilitated knowledge transfer related to implementing the pain protocol. They said that the clinical nurse specialist and nurse practitioners were dedicated to the topic and positive about the change, which facilitated buy-in and motivated staff*”.[45]

#### 3.2.3. Facilitating Leadership Capability within Teams

##### Providing Autonomy; Encouraging Staff to Apply Their Knowledge, Skills and Capacity

The BN/MN facilitated in offering the team(s) room to grow and to be the owner of the change-in-care processes, and then to tackle these independently. Each team member’s talent was used to give him or her the opportunity to grow and to gain control over [implementation or change of] care.

The BN/MN gave the team a sense of confidence and self-confidence by encouraging members to continue to fulfill the task and grow in their roles as future leaders. The BN/MN contributed elements important to the team’s self-confidence such as safety, trust, well-being, and self-respect.

“*The director of nursing [DON] can provide the vision to inspire the rest of the team, model the appropriate behavior, and encourage others to take leadership positions*”.[49]

“*Celebrating successful improvements reinforces doing the right thing and encourages staff to continue and sustain the improvement processes*”.[49]

“*…she usually tries to find me [licensed nurse] throughout the building……..”. And it gives you a good feeling too because she’s [MN] also building the morale staff wise as well. She’s giving you that confidence to go on and continue*”.[50]

##### Facilitating Conditions for Team Performance

The BN/MN facilitated team(s) in leadership capacities and therefore contributed to the leadership quality of a team by creating capacity and opportunities within the organization. The BN/MN facilitated team members to grow and develop in their role, inspired them, and gave them opportunities by deploying various strategies.

“*At the ward level, they [board members of organizations] expected BNs to fulfil an informal, clinical leadership role for direct care staff and helping teams to implement care innovations*”.[46]

“*The director of nursing [DON] can work with the nursing home administrator [bachelor or master educated] to ensure that the quality improvement team [registered nurses] receives the management support and resources necessary to enable success*”.[49]

#### 3.2.4. Facilitating Implementation of Guidelines, Standards, and Protocols

##### Facilitating the Implementation of Evidence-Based Quality Improvement

Working in close collaboration, the BNs/MNs facilitated teams to make an active contribution to improvement and change processes in elderly health care. The emphasis was on the successful implementation of protocols, standards, and/or guidelines, and on monitoring, guiding, and facilitating the process towards implementation. The BN/MN monitored quality of care by conducting audits and supplying feedback, as a way of encouraging the teams to continue with what they were doing well and to maintain the change or to bring attention to domains that required refinement or quality improvement.

“*As a registered nurse leader, the director of nursing [DON] can provide functional management and oversight of the quality improvement project by assigning appropriate staff to participate in quality improvement as well as identify priority areas that need improvement*”.[49]

“*The clinical nurses and nurse practitioner used a variety of strategies to help implement the pain protocol in long-term care. They were seen as change champions and active in organizing and facilitating interdisciplinary practice to reinforce the pain protocol and provide ’check-ins’ with staff to identify barriers to implementation*”.[45]

“*The clinical nurses and nurse practitioners were responsible for organizing and facilitating monthly interdisciplinary pain team meetings with staff to help implement the pain protocol and problem solve issues together*”.[45]

##### Facilitating Various Strategies to Maintain Improvement

The BN/MN made sure that procedures were evidence-based and that teams received adequate training and education on integrating quality in the primary process.

“*For the quality improvement team, the director of staff development [DSD] can help to identify evidence-based practices and provide staff education about the particular quality improvement intervention, depending on their [quality improvement team] licensure and skill set*”[49]

“*The nurse practitioner developed a case study (for discussion at a pain team meeting) to work through pain protocol and enhance application of knowledge and problem-solving ability*”.[45]

#### 3.2.5. Facilitating a Work Environment to Acknowledge Grief and Loss of Residents within Teams

Nursing home care providers face many things, including the death of clients, which may be either expected or sudden. Like family members, health-care providers experience grief as well. Grief is the personal emotional response to the experience of loss. Bereavement is the experience of the death of a significant loved one. The BN/MN provided peer support when teams experienced grief after losing a client.

The BN/MN encouraged team members to get together with other team members and reminisce following the death of a client. This provided team members with an opportunity for personal closure.

“*NPs also provide support to other LTC staff in terms of their own bereavement when a resident dies. An administrator stated that the NP ‘spends time with the staff in debriefing, and works with the staff’ to help them come to terms with the loss, particularly when the death was difficult*”.[50]

“*…the Reflective debriefing groups [led by a nurse specialist] fulfilled both a supportive and communicative role among team members. Many staff benefited from being able to open up about personal losses*”.[43]

## 4. Discussion

The aim of this integrative review was to identify what competencies are required of bachelor- or master-educated nurses in facilitating the development of an effective workplace culture in nursing homes. We identified five core competencies: (1) facilitating a learning culture in nursing practice; (2) facilitating effective work relationships within teams; (3) facilitating leadership capability within teams; (4) facilitating the implementation of guidelines, standards, and protocols; and (5) facilitating a work environment to acknowledge grief and loss of residents within teams.

These five core competencies show the skills and qualities of a BN/MN in a facilitator role as they support developing nursing practice within and collaboratively with teams. According to Manley and Jackson [27], facilitators use the workplace as the main resource for learning, development, improvement, knowledge translation, and innovation [27]. In our results, the BN/MN demonstrated these facilitating activities in the nursing homes. However, the scope and range of facilitating individuals and groups through organizations and systems—with the primary intent of enhancing higher quality of care and professionalization of staff—was not fully clear. From our perspective, developing practice while practicing within an organizational and system context requires leadership that facilitates empowerment of frontline staff and their teams to sustain person-centered practice. It is essential that teams experience leadership from within, rather than from above. The facilitation then leads to greater collaboration and commitment from teams.

In order to develop sustainable person-centered practice, McCormack et al. [16] suggest that clinical and professional leadership by BNs/MNs is needed to facilitate feelings of satisfaction and involvement in teams, and feelings of well-being for residents and nurses [16,59]. According to Manley et al. [26], transformational leadership is the most influential leadership style in effective workplace cultures [26]. According to Martin et al. [60], a shift is needed from hierarchical leadership to a leadership style encouraging support structures to facilitate staff empowerment. Transformational leadership has a high impact on practice change in nursing and on the development of an organizational structure [60]. This leadership style is characterized by the ability to stimulate, inspire, and motivate teams [61].

In our results, the BN/MN emerges as a leader who connects individuals and team members within their work environment and invests in self-confidence and team efficacy. Furthermore, the BN/MN facilitates autonomy by encouraging team members to apply their knowledge, skills, and capacities, helps others to reach their highest potential, and remains committed to the development of others. Successful leadership by BNs/MNs can improve the performance of individual nurses and team members in achieving common goals within an organization. In essence, the BN/MN stimulates individual nurses to carry out their tasks independently, without guidance from a BN/MN. In the context of transformational leadership [62], the BN/MN deploys competencies such as facilitating interactive decision sharing and responsibility-sharing between individual nurses and team members, and empowering and motivating staff within developmental processes. McCormack et al. [16] emphasize this leadership style where shared power and shared decision-making are the relationships that take place in the care environment [16,63]. This implies that BNs/MNs are encouraged to support teams to make bottom-up decisions and drive innovative behavior towards improving nursing practice, rather than taking a top-down approach.

However, being in the facilitator role does not necessarily mean that one is a leader; nonetheless, a leader can certainly be a facilitator of (culture) change. It is not clear whether the BN/MN uses transformational leadership effectively by the ability to create change in team member thinking and perception, thus ensuring person-centered safe and effective care. On the other hand, the results show that the BN/MN creates a work environment to support developmental processes, but not with the intention of transforming culture from a shared organizational purpose and vision. It is important to have leadership at all levels in the organization to maintain change on all levels within the organization [64]. Cardiff et al. [25] describe several additional competencies this requires, such as credibility, networking skills, inter- and intrapersonal skills, and leadership qualities [26]. Our results do not expressly point out these skills and qualities, with the possible exception of leadership qualities. Leadership at all levels requires the development of change champions throughout the organization who are skilled in initiating, facilitating, and implementing change [65]. Leadership and change champion from a facilitating role is the most overarching competency in this review. More research is needed to explore the role and enculturation of transformational leaders in the nursing home.

Most of the studies described the facilitative role of the BNs/MNs with an intent to enhance quality improvement and innovation in older people care. The BNs/MNs acted as a sort of knowledge broker in promoting evidence-based practice among team members. The BNs/MNs collected data, performed audits, and held sessions to evaluate whether changes in process and practice helped attain desired results. In addition, the BNs/MNs used a variety of implementation strategies that were evidence-based, such as a pain protocol. This is in line with both the study of Gerrish et al. [66] on BNs/MNs who act as change champions to facilitate the adaption of best practice to respond to challenging and dynamic practice contexts [66], and evidence-based quality improvement [EBQI] studies about implementing research evidence into practice in a structured way [67,68,69].

We found that the BNs/MNs also played a pivotal role in the well-being of individual nurses and team members, particularly during the grieving process after losing residents. The teams desired this form of support, and it offered them comfort in dealing with the emotional challenges of working in a long-term care environment. In other studies, staff identified peer support as the most beneficial level of support [70]. As part of person-centered care, it is essential that, as facilitators, BNs/MNs recognize potential complications of grief to foster healthy relationships with teams. The BN/MN can create a safe and trustworthy work environment in which teams are allowed to express their emotions and show vulnerability. Supporting grief and loss in the work environment/person-centered care environment is needed, and it contributes to higher staff (job) satisfaction and ultimately retention [22,70,71]. In our results, BNs/MNs demonstrated support by initiating bereavement debriefing sessions as a strategy to address staff retention and satisfaction. In Keene et al., [72] the debriefing session is one example of support that an organization can provide as part of a multifaceted approach in support for its staff [72]. In addition, the use of debriefing sessions is an essential method to reflect through experiential learning [73,74]. However, individual nurses and teams are also exposed to personal matters related to work–life balance. High stress levels can affect a nurse’s health and well-being, which has a negative impact on the work environment. The BNs/MNs can also play a role in minimizing stress by supporting teams to enhance function when facing workplace challenges [75].

In our study, the BN/MN deploys strategies to initiate activities to facilitate an effective workplace culture, but it was not clear there was deliberate intent to change or transform workplace culture to become more person-centered. This study suggests investing in leadership and facilitation skills, but it does not specify how to do so or whose task or responsibility this is. Furthermore, the study revealed that the BNs/MNs are well-placed to promote person-centered and evidence-based practice through an effective collaboration with individual nurses and team members in nursing homes. However, it was not always clear whether the competencies we identified as contributing to an effective workplace culture were displayed by BNs/MNs deliberately. It was also not always clear whether the intent was to contribute to workplace culture or [just] to encourage team members to be responsible and undertake continuous professional development.


**Study Limitations**


This review has some strengths and limitations. First, we found only a small number of studies regarding workplace cultures in nursing homes. A more substantial number of studies is needed to study cultural changes/transformations and differences in nursing homes. Second, it was not always clear in the articles which nursing staff were included. The description or grades of participating nursing staff were either not described or described insufficiently. Furthermore, it was not clear whether the master-educated nurses were nurse practitioners or nurses with an academic/nursing science degree. During our search, we retrieved two articles focusing on “registered nurses”. Third, the 16 studies consisted of varying study designs and quality. Most of the studies originated from Canada, where the nursing-home-work environment is not quite comparable [generalizability] to the situation in other countries. Fourth, the studies were appraised as relevant and useful. The CASP tools were used in the quality appraisal [42]. In two studies there were some quality shortcomings.

In this review, in addition to searching for articles that focus on the BNs/MNs’ professional competencies, we also performed an extensive search using synonyms of competencies (e.g., “skills”, “knowledge”, and “attitude”). The combined results of this search provided a comprehensive and rewarding overview.

The study was strengthened by the repeated reviews of two independent peer reviewers. The first author assessed all articles. Two of the co-authors reviewed a selection of the studies. At each phase of the selection process, the results that emerged were discussed, defined, and refined by the two co-authors in three meetings. Most of the articles were qualitative/mixed-methods studies, which gives an in-depth insight into the context of nursing practice and shared values and beliefs in everyday work and practice.

Despite its limitations, we are convinced that the results of this review provide a good first impression of the advanced competencies required of BNs/MNs in facilitating an effective workplace culture in nursing homes.


**Implications**


Further research is needed into the characteristics of a positive work environment in older people care and the role of workplace culture in this environment. Most of the studies (10 out of 16) included master-educated nurses. It may be useful to study the specific role of bachelor-educated nurses in a function mix. It would also be interesting to explore changing the staff and skill mix of health-care teams for improving the quality of care for the older people and organizational effectiveness. Further research is needed to study differences of education (advanced degrees or specialty), expertise, and competencies between employees who work in the same health-care team. In addition, it is also interesting to analyze the nursing competencies within and between countries and institutions regarding the role of workplace culture to improve the transparency of nursing qualifications, skills, training, experience, and work environment throughout the global healthcare system.

Furthermore, it is essential to study the experiences of BNs/MNs regarding their role in enabling an effective workplace culture, and, in addition, what role hierarchical leaders such as managers and directors can play in practice development.

We suggest that future research validates the competencies identified as facilitating an effective workplace culture in nursing homes. Bachelor’s and master’s programs can consider adding these advanced competencies to their curricula. Furthermore, they can develop a new training profile for the Bachelor of Nursing and the Master of Advanced Nursing Practice.

By organizing nurse leaders on all levels within a health-care organization and investing in the development of leadership and facilitation skills, nurses can and should be supported to ensure and maintain safe and effective person-centered practice. The results show the complexity of describing the role of the facilitator enabling facilitation within the workplace culture. It is not always clear which key attributes and skills are needed to facilitate workplace culture (change). Most studies describe the role of a champion who fosters and reinforces changes for improvement.

## 5. Conclusions

This review identified five core competencies for BNs/MNs to facilitate the development of an effective workplace culture in nursing homes. It shows that the BN/MN applies competencies associated with a facilitator role to promote the development of a person-centered workplace culture and thereby achieve safe, high-quality care, satisfaction, and well-being. Overarching aspects include: exhibiting leadership to be a change champion recognizing the need for change, creating a new vison, and institutionalizing change in practice in collaboration with teams. This integrative review indicates that advanced leadership competencies are required to achieve transformation across frontline staff and teams.

## Figures and Tables

**Figure 1 ijerph-19-12324-f001:**
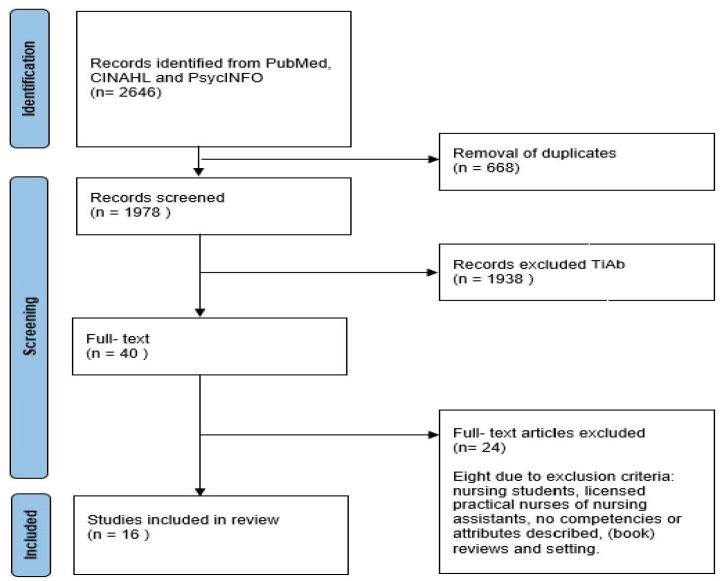
Flow Diagram showing the article selection process [41].

**Figure 2 ijerph-19-12324-f002:**
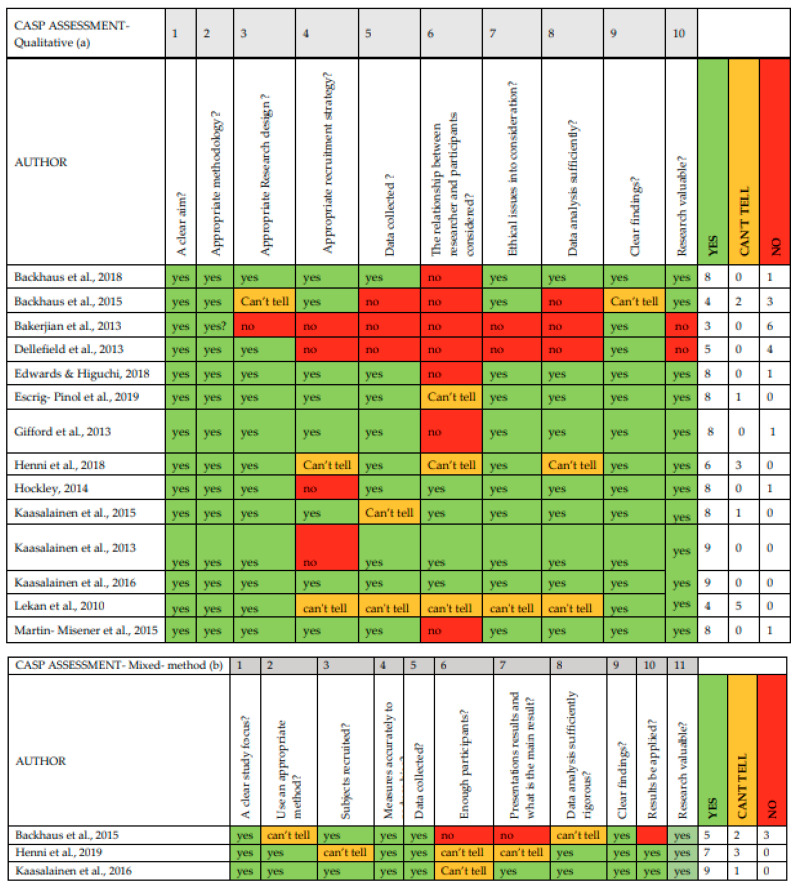
CASP Assessment [42], summary review of the assessment of the included studies [43,44,45,46,47,48,49,50,51,52,53,54,55,56,57]. Green-coded box indicates that the study satisfactorily met the respective quality criterion, yellow-coded box indicates that the study partially met the respective quality criterion, and the red-coded box indicates that the study did not meet the respective quality criterion.

**Table 1 ijerph-19-12324-t001:** Inclusion and Exclusion Criteria.

Inclusion	Exclusion
Population: bachelor-educated nurses, master-educated nurses in nursing science, MSN nurse practitioners, Master Advanced Nursing Practice (MANP). Setting: nursing homes, intramural long-term elderly care. Design: mixed-methods designs, qualitative studies, quantitative studies, observational studies, before-and-after studies, implementation studies, expert opinion (nationally and internationally known, conducted high-quality research, published in peer-reviewed journals). Focus: professional nursing competencies (knowledge, skills, expertise, capacity, role of coaching/facilitating nursing teams). Domain: UK, Australia, New Zealand, Canada, United States, and Europe (such as the Netherlands, Germany, Switzerland, and Scandinavian countries). Publication date: <10 years (Limits: 2010–2020). Outcome: workplace culture or workplace facilitation of learning.	Population: nursing students. Population: education other than bachelor’s degrees; licensed practical nurses, nursing assistants, nursing managers, district nurses, supervisory nurse, physician assistant. Setting: hospital care/critical care, home care, psychiatric care, public health, community health, primary care, outreach service, rehabilitation setting, hospice, adult home care, chief nursing officer CNO). Language: other than English or Dutch. Design: (book) reviews, studies not related to nursing home contexts, studies not related to the bachelor-educated nurses’ work and studies not reflecting the Bachelor of Nursing competencies needed for older people care. Domain: non-western/countries not listed in the inclusion domain. Outcome: person-centered care or patient-centered care, outcomes in direct care.

**Table 2 ijerph-19-12324-t002:** Characteristics of included studies.

No.	Author(s), Year, Country	Title	Aim Study	Object of Study	Methodology	Participants
**1**	Backhaus et al., 2018 [46]; the Netherlands	Baccalaureate-educated Registered Nurses in nursing homes: Experiences and opinions of administrators and nursing staff	To understand how nursing homes employed BRNs and how they viewed the unique contributions of BRNs for staff and residents in their organizations.	Bachelor registered nurse, EQF 6.	- A qualitative study. - Semi-structured, face-to-face individual or group interviews. - 6 NH.	Board members, directors, ward/nursing home managers, and BRNs: N = 26 individual interviews and 3 group interviews (N = 14).
**2**	Backhaus et al., 2015 [51]; the Netherlands	Future distinguishing competencies of baccalaureate-educated registered nurses in nursing homes	To obtain insight into the competencies, which should in the future distinguish BRNs from other nursing staff in nursing homes.	Bachelor registered nurse, EQF 6.	- Expert consensus study. - Digital survey followed up by expert panel to discuss the survey results, and a final survey to determine the degree of consensus.	N = 41 experts (no BNs were included) from various countries.
**3**	Bakerjian & Zisberg, 2013 [49]; the U.S.	Applying the Advancing Excellence in America’s Nursing Homes Circle of Success to improving and sustaining quality	To describe the potential roles and responsibilities of registered nurse leaders in NHs in implementing and sustaining a comprehensive improvement program.	Nursing leaders: RN’s and director of nursing (DON) and director of staff development (DSD), EQF 4–6.	- Quality improvement process. - Single case study of 1 NH.	1 DON; 1 DSD; 1 team leader; 1 APN from a local university; 1 quality improvement (QI) consultant.
**4**	Dellefield et al., 2013 [52]; the U.S.	Quality assurance and performance improvement in nursing homes: Using evidence-based protocols to observe nursing care processes in real time.	To describe quality assurance and performance improvement (QAPI) elements and explain how to collect data using direct observation and evidence-based measures and protocols in a QAPI program.	Registered nurses, EQF unknown.	- Implementation study. - Direct observation; quantitative data collection used for PI programs. - Evidence-based measures and protocols to collect data on staff member actual performance of care processes.	Registered nurses and other nursing staff (Nursing assistants).
**5**	Edwards & Smith Higuchi, 2018 [53]; Canada	Process evaluation of a participatory, multimodal intervention to improve evidence-based care in long-term care settings	To examine the effect of a multimodal, participatory intervention aimed at improving evidence-based care for the residents of LTC homes.	Best practice Coordinator (BPC); master or bachelor degree. Nurse external facilitator. EQF: 6–7.	- Process evaluation. - Qualitative descriptive design. - Semi-structured individual interviews conducted by telephone. - 12 sites.	Staff members: registered and non-registered staff and other health members. Different disciplines, levels, and departments of the LTC sites. External facilitator, nurse facilitator, and best practice coordinator. N = 44 at midpoint and N = 69 at endpoint.
**6**	Escrig-Pinol et al., 2019 [47]; Canada	Supervisory relationships in long-term care facilities: A comparative case study of two facilities using complexity science	To seek a better understanding of the factors that contribute to effective supervisory performance in LTCFs.	Nurse supervisor (RN or RPN) as participants, EQF: 6	- Comparative case study. - Semi-structured interview.	Two LTCFs: Case-study 1: N = 10 Case-study 2: N = 10 including management (n = 6) (administrators, directors of care and middle managers staff), RNs, and RPNs (n = 7) and PSWs (n = 7).
**7**	Henni et al., 2018 [54]; Norway	The role of advanced geriatric nurses in Norway: A descriptive exploratory study	To describe the experience of nurses with their new role as AGNs in care of older adults, and to determine what strategies the AGNs consider important in the development of their new role.	Advanced Geriatric Nurses, EQF: 7	- Descriptive/exploratory design. - In-depth interviews.	21 AGNs graduated from the University of Oslo before summer 2016.
**8**	Henni et al., 2019 [55]; Norway	The integration of new nurse practitioners into care of older adults: A survey study	To investigate the level of integration of AGNs in their fields of practice; use of their knowledge and skills to reach their full potential.	Advanced Geriatric Nurses, EQF: 7	- Cross-sectional descriptive survey. - A questionnaire.	AGNs (n = 26) and colleagues (n = 465).
**9**	Hockley, 2014 [43]; the U.K.	Learning, support and communication for staff in care homes: outcomes of reflective debriefing groups in two care homes to enhance end-of-life care	1. To identify problems that staff experience in caring for a resident who is dying and the impact on the provision of high-quality end-of-life. 2. To examine what actions could successfully be implemented in order to promote high-quality end-of-life care.	Nurse specialist in palliative care, as the leading facilitator, EQF: 7	- Action research design: to explore and develop quality end-of-life care in two nursing homes. - Reflective debriefing groups and evaluation questionnaire. - 2 NHs.	10 RdBGs, per RdBGs; 34 different staff members attended RdBGs; nurse managers, nurses, and health care assistants (HCA).
**10**	Kaasalainen et al., 2015 [45]; Canada	Positioning clinical nurse specialists and nurse practitioners as change champions to implement a pain protocol in long-term care	To explore the role of a clinical nurse specialist and a nurse practitioner as change champions during the implementation of an evidence-based pain protocol in LTC.	Advanced practice nurses: Nurse practitioners (NP) and Clinical nurse specialist (CNS), EQF: 7	- Exploratory multiple-case study design. - 2 LTC facilities: 1 site employed a NP and site 2 employed a CNS.	1. APN diaries (first 3 months) completed by CNS and NP. 2. Participant observation (36 h): research assistant shadowed each CNS and NP. 3. Four focus group at the end of implementation phase: 2 with PSWs (n = 17) and 2 with RPNs and RNs (N = 11), Individual interviews: members of administration (N = 5) and interdisciplinary team members (N = 4) and the NP and CNS.
**11**	Kaasalainen et al., 2013 [50]; Canada	Role of the nurse practitioner in providing palliative care in long-term care homes	To present study findings about the role of the NP in providing palliative care in long-term care homes (LTCs).	Nurse practitioners, EQF: 7	- Descriptive design. - Five LTC homes that employed an NP (two homes shared one NP).	Physicians (n = 9), licensed nurses (n = 20), PSW or healthcare aides (n = 15), managers (n = 19), RN managers or leaders (n = 10), allied health care providers (n = 31), NPs (n = 4), residents (n = 14), and family members (n = 21). N = 143 were interviewed individually or in a focus group; 35 focus groups and 25 individual interviews.
**12**	Kaasalainen et al., 2016 [56]; Canada	The effectiveness of a nurse practitioner-led pain management team in long-term care: A mixed methods study	1. To evaluate an NP-led, IP pain management team in LTC. 2. To evaluate the effectiveness of the implementation of the NP-led pain management team in improving resident outcomes and health-care provider outcomes. 3. To explore staff perceptions of the implementation of the NP-led pain management team.	Nurse practitioners, EQF: 7	- a mixed method design. - controlled before–after study. - 6 LTC homes were allocated to one of three groups. Each group included 2 LTC homes; medium-to-large-sized facility, employment of an NP and use of pain management team.	Total: 139 residents full intervention group; 108 residents to partial intervention group and 98 residents in control group.
**13**	Lekan et al., 2010 [44]; U.S.	The Connected Learning Model for disseminating evidence-based care practices in clinical settings	To describe the development, implementation and feasibility evaluation of the Connected Learning Model to facilitate adoption of heart failure clinical practice guidelines for symptom recognition in one nursing home.	Advance practice nurse as the facilitator, EQF: 7	- Implementation design. - 114-bed nursing home.	Four nursing units staffed with: RN, LPN, NA nursing staff and RN supervisor (n = unknown). Evaluation of the feasibility of different teaching–learning strategies and on-the-job skill competency evaluation to verify learning.
**14**	Martin-Misener et al., 2015 [48]; Canada	A mixed methods study of the work patterns of full-time nurse practitioners in nursing	To explore the integration of the NP role in Canadian nursing home settings to enable the full potential of this role to be realised for resident and family care.	Nurse practitioner, EQF: 7	- A mixed-method study. - Cross-sectional national survey followed by four case studies in NH; individual and focus group interviews.	Survey: Nurse practitioners (N = 26). Case-study: healthcare providers, administrators, family members, multidisciplinary health professionals, NPs, managers, physicians, residents, and unregulated care staff were interviewed via individual interview or focus group (N = 150).
**15**	McGilton et al., 2016 [12]; International	Recommendations From the International Consortium on Professional Nursing Practice in Long-Term Care Homes	1. To describe recommendations about priority issues for action and a research agenda regarding the RN in LTCHs. 2. To reach consensus on priority issues future research.	Registered nurses, EQF: unknown.	- A consensus study. - Two days meeting.	Consortium: Nursing experts engaged in research, policy, administration/operations, and education in aging and LTC.
**16**	Gifford et al., 2013 [57]; Canada	Moving Knowledge to Action: A Qualitative Study of the Registered Nurses’ Association of Ontario Advanced Clinical Practice Fellowship Program	To describe the perceptions of Advanced Clinical Practice Fellowship (ACPF) fellows regarding their influence on quality of care and patient outcomes through advanced nursing knowledge translation and skills development.	Masters prepared nurses, EQF: 7	- Survey. - Telephone interviews (n = 31).	Primary mentors of the ACPF fellows.

## Data Availability

Not applicable.

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
