# Peer review of "The Required Competencies of Bachelor- and Master-Educated Nurses in Facilitating the Development of an Effective Workplace Culture in Nursing Homes: An Integrative Review"

_ijerph, 2022, doi:10.3390/ijerph191912324_

Round 1
Reviewer 1 Report
Thank you for the opportunity to review the manuscript titled: The Required Competencies of Bachelor- and Master-Educated Nurses in Facilitating the Development of an Effective Workplace Culture in Nursing Homes: An Integrative Review.
The culture of the healthcare workplace is influential in delivering person-centered care, clinically effective, and continually improving in response to a changing context. From the literature review, we know that nursing management and leaders must take in consideration that works culture is crucial for improving the quality of care in nursing homes. Also, the management should focus on the work culture among healthcare personnel in nursing homes.
Below are my comments:
1. Please specify the date range of the research: in the method section October 2021 to December 2021; in the Materials and Methods section, The search was limited to studies published between January 2010 and
May 2020 in English and The search was conducted between October 2021 and December 2021.
2. In the main text, there is a lack of tables 2 and 3
3. In the introduction section, please specify the recommendation for Nurse Competencies for Nursing Home Culture.
4. Did you analyze the nursing competencies across the countries, and did you compare the competencies for the BN and MN level of education
Reviewer 2 Report
First of all, congratulations for the work done, it is a very interesting work. The work to be reviewed has an adequate introduction, justification of the contents to be studied as well as the results. It is essential for the publication of this article to improve the methodology section. After reviewing the manuscript, it could be improved in the following points:
It is necessary to better justify the choice of an integrative review as the research methodology.
The authors should justify the reason for limiting the study to 10 years, from 2010 to 2020.
What is the reason for leaving countries such as France, Spain or Italy out of the study?
It is necessary to include as supplementary material the search strategy performed.
Kind regards
Round 2
Reviewer 2 Report
The article has been substantially improved by the changes suggested in the first revision. Congratulations on the work.
Best regards.